# A 14 × 14 μm² footprint polarization-encoded quantum controlled-NOT gate based on hybrid waveguide

S.M. Wang[1,2,*], Q.Q. Cheng[1,2,*], Y.X. Gong[3,*], P. Xu[1,2], C. Sun[4], L. Li[1,2], T. Li[1,2] & S.N. Zhu[1,2]

Photonic quantum information processing system has been widely used in communication, metrology and lithography. The recent emphasis on the miniaturized photonic platform is thus motivated by the urgent need for realizing large-scale information processing and computing. Although the integrated quantum logic gates and quantum algorithms based on path encoding have been successfully demonstrated, the technology for handling another commonly used polarization-encoded qubits has yet to be fully developed. Here, we show the implementation of a polarization-dependent beam-splitter in the hybrid waveguide system. With precisely design, the polarization-encoded controlled-NOT gate can be implemented using only single such polarization-dependent beam-splitter with the significant size reduction of the overall device footprint to 14 × 14 μm². The experimental demonstration of the highly integrated controlled-NOT gate sets the stage to develop large-scale quantum information processing system. Our hybrid design also establishes the new capabilities in controlling the polarization modes in integrated photonic circuits.

[1] National Laboratory of Solid State Microstructures, School of Physics, College of Engineering and Applied Sciences, Nanjing University, Nanjing 210093, China. [2] Collaborative Innovation Center of Advanced Microstructures, Nanjing 210093, China. [3] Department of Physics, Southeast University, Nanjing 211189, China. [4] Department of Mechanical Engineering, Northwestern University, Evanston, Illinois 60208-3111, USA. * These authors contributed equally to this work. Correspondence and requests for materials should be addressed to S.M.W. (email: wangshming@nju.edu.cn) or to T.L. (email: taoli@nju.edu.cn) or to S.N.Z. (email: zhusn@nju.edu.cn).

The quantum controlled-NOT (CNOT) gate is one of the fundamental building block for quantum information system that flips the target qubit state conditional on the control qubit being in the state $|1\rangle$. It enables the construction of any quantum computing circuits when combined with single-qubit gates[1]. A widely used linear optical CNOT gate[2,3] manipulating path qubits by path interference with multiple beam-splitters has been first realized by using free-space optical components[4] and subsequently using integrated waveguides[5]. By employing the polarization-encoded scheme, the gate can be significantly simplified by requiring only three polarization-dependent beam-splitters (PDBSs) and thus, results in further size reduction and improved stability by eliminating the phase-sensitive interference[6–8]. Recently, such simplified CNOT gate has been realized using the femtosecond-laser-written directional couplers with the precise control on the splitting ratio for the orthogonally polarized modes. The use of directional couplers requires quite long interaction length and thus, the overall dimension of the gate remains in a millimetre scale in order to gain independent control of the orthogonally polarized modes[9]. Hence, a more compact CNOT gate is yet to be explored for addressing the urgent needs in developing large-scale quantum information processing system.

In the following, we report the implementation of a hybrid PDBS by strategically combining the dielectric and plasmonic waveguide that each is dedicated to handle the TE (transverse electric) and TM (transverse magnetic) polarized mode all within the same component. With precisely design of the output slits, the polarization-encoded CNOT gate can be implemented using single such PDBS with the significant size reduction of the overall device footprint to $14 \times 14 \, \mu m^2$. The gate demonstrates the good quantum CNOT functionality with a high fidelity.

## Results

**Classical characterization of the HW-based PDBS.** The polarization-encoded CNOT gate[6–8] is schematically shown in Fig. 1a. The core part of the gate is a PDBS (PDBS$_0$) that allows 100% transmission of the TE polarized light and $\xi/3$ ($2\xi/3$) transmission (reflection) of the TM polarized light, where $\xi$ is the total coefficient of the system. Auxiliary PDBSs (PDBSa), with transmittances (T) obeying $T_{TE}/T_{TM} = 1:2$, are employed to balance the contributions from the two polarizations. We have employed the design of a hybrid waveguide (HW), namely dielectric-loaded surface plasmon polariton (SPP) waveguide, which supports both TM (SPP) and TE (photon) modes[10]. The SPPs have been proven to be a valid carrier of quantum information[11,12]. Recently, experiments have further verified the bosonic nature of SPPs via an on-chip non-classical interference[13–17].

Our HW-based PDBS is presented in Fig. 1b. The HW system is comprised of a 300-nm silver film covered with a 250-nm silica film, which is chosen to ensure that only the lowest TM mode (SPP) and the lowest TE mode are supported in the system (see Supplementary Fig. 1 and Supplementary Note 1 for details). The input couplers, gratings milled on the silver film by focused ion beam etching (Helios nanolab 600i, FEI), are utilized to couple the incident light into the waveguide modes. The p-polarized light with the electric field perpendicular to the gratings is coupled to the SPPs supported TM mode in the HW system, whereas the s-polarized light with the electric field along the gratings is coupled to the TE mode. The excited SPPs, with excellent directionality and beam quality[18], propagate along the metal surface. A 45° rotated grating milled on the silver surface 4.5 μm away from the input coupler works as an SPP beam-splitter whose transmission and reflectance can be conveniently tailored

by controlling its period, width and depth. For the convenience of extraction and measurement of the quantum information carried by the polarization-encoded waveguide modes, output couplers, five slits, located 4.5 μm away from the SPP beam-splitter are used to convert SPPs back into p-polarized light on the back side of the device. A scanning electron microscopy image of the sample configured on the surface of silver film is presented in Fig. 1c. The output light is collected by a leaky mode microscopy system[10,18], in which an oil-immersed objective lens is used. The optical images of the back side of the device, with a p-polarized 785 nm laser illuminating each of the two input couplers, are shown in Fig. 1d,e, respectively. After carefully tuning parameters of the SPP beam-splitter, the PDBS can be fabricated with the measured transmittance/reflectance (T/R) ratios being 1:1.9 and 1:2.15, respectively, for different input gratings, commensurate with the ideal ratio 1:2 required (see Supplementary Fig. 2 and Supplementary Note 2 for details). The deviation between these T/R ratios may be due to the fabrication imperfections during focused ion beam etching.

On the other hand, the HW system also supports the lowest TE mode that can be excited by s-polarized light by sharing the same input couplers for SPPs. Owing to the obvious difference in wavelengths and field distributions between the TE mode and SPP, the SPP beam-splitter has little influence on the TE mode. This leads to a nearly unity transmittance of TE mode at the SPP beam-splitter, which satisfies the perfect transmission required in

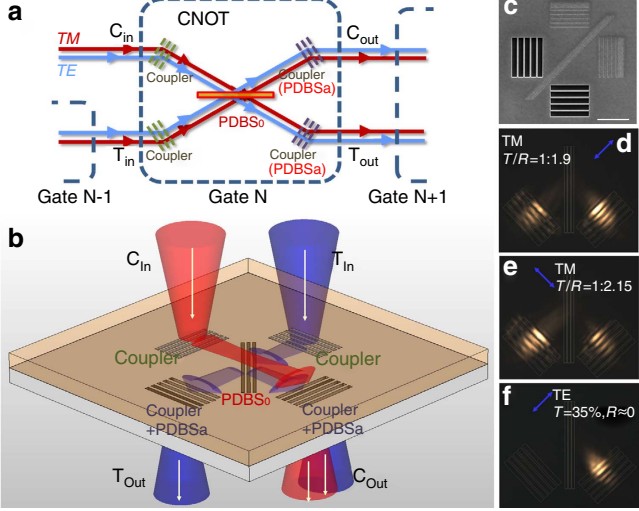

**Figure 1 | Classical characterization of the HW-based CNOT gate.** (**a**) A schematic of the simplified CNOT gate composed of three PDBSs as proposed in refs [7] and [8]. The photons enter the gate from the input couplers C$_{in}$ (control) and T$_{in}$ (target) and get out from the output couplers C$_{out}$ and T$_{out}$. (**b**) The sketch of the CNOT gate realized by the HW-based PDBS. (**c**) The scanning electron microscopy image of the PDBS on a metal film. The input coupler consists of two kinds of gratings: the working gratings having five grooves with depth $d = 30$ nm, length $l = 5 \, \mu m$, width $w = 200$ nm and period $P = 540$ nm, and the reflecting gratings with different specifics, $w = 100$ nm and $P = 270$ nm for high conversion efficiency. This approach aims to obtaining the high conversion efficiency from p-polarized light to SPP, in a direction-dependent way. The SPP reflecting grating comprises of three grooves with $d = 50$ nm, $w = 100$ nm and $P = 380$ nm. The slits of the output coupler have $l = 5 \, \mu m$, $w = 600$ nm and $P = 830$ nm. The scale bar denotes 4 μm. (**d,e**) The CCD images of the output setup with p-polarized light input from the left coupler (**d**) p-Polarized light input from the right coupler (**e**) and s-polarized light input from the left coupler (**f**) where the blue arrows mark the positions of input lights and their polarizations.

the CNOT gate. In this way, one can simultaneously manipulate the two polarized waveguide modes using the HW system. Moreover, the five slits of the output couplers not only can convert the TE mode back to s-polarized light for collection but also can be used to accurately tune the intensity of TE mode to $1/3\xi$ by precisely adjusting their parameters for balancing the contributions of the TE mode and SPP in the gate. Therefore, the two auxiliary PDBSs for TE mode attenuation used in the previously scheme can be removed and the architecture of the polarization-encoded CNOT gate based on HW system is further simplified to a single PDBS. As a result, in this work, the footprint of the polarization-encoded CNOT gate is reduced to only $14 \times 14\,\mu m^2$, which is two orders smaller than implementations in dielectric waveguide system[5,9] and is much promising for future quantum photonic integration. The optical image of the back side of the device with s-polarized light illuminating on one input coupler is shown in Fig. 1f. The transmittances are $37\%\xi$ and $35\%\xi$ for input from the right and left gratings, respectively. Nearly zero reflection of the TE mode can be observed in both cases. As the results for the two input gratings are similar, here without loss of generality we only present the case of left grating input. The 45° alignment of the gate is employed for the convenience of separately collecting the photons at the two outputs.

**The quantum characterization of the HW-based CNOT gate.** The experimental setup is sketched in Fig. 2. A pulsed 785 nm femtosecond laser from a Ti: Sapphire oscillator is frequency-doubled using a 2-mm-thick β-barium borate (BBO) crystal to obtain 392.5 nm pulses. These pulses then pump a 2-mm-thick type-II BBO crystal for generating collinear photon pairs with orthogonal polarizations. The quality of the photon-pair source can be characterized by using the two-photon Hong–Ou–Mandel (HOM) interference[19]. The coincidence of the HOM interference of the photon-pair source is shown in Fig. 3a. The HOM dip with high visibility $V = (C_{max} - C_{min})/C_{max} = 95.2 \pm 0.7\%$, manifests its good quality for further quantum interference utilization[13–17]. We then use this photon-pair source to characterize the PDBS we fabricated in the HW system. The orthogonally polarized photons, separated by a polarizing beam splitter, are collected with single-mode fibres and injected into the HW-based PDBS to excite SPPs with their polarizations both rotated to the p-polarization using the polarization controller 1. The excited SPPs interfere at the PDBS, with the best interference effect obtained by carefully controlling the temporal overlap in the delay line. The output SPPs are then coupled to the propagating wave by the output couplers and subsequently collected using an oil-immersed objective lens. The HOM interference of the SPPs on this HW-based PDBS is depicted in Fig. 3b. The visibility of $72.4 \pm 3.1\%$ has been observed, which is close to the theoretical value of 80% for an ideal 1:2 beam-splitter. The result also proves the bosonic nature of SPPs and demonstrates the good quality of SPPs as a quantum information carrier[13–17].

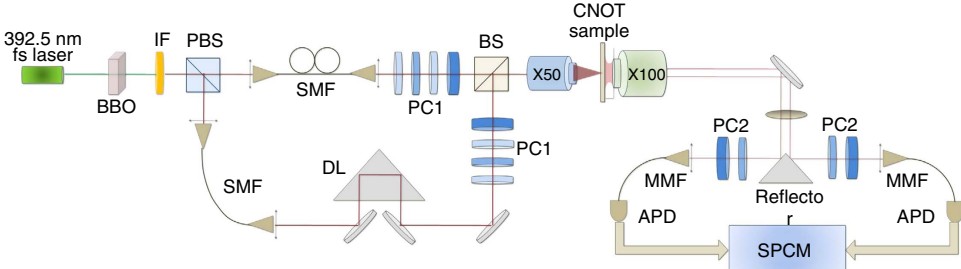

**Figure 2 | The experimental setup.** The photon pairs at wavelength $\lambda = 785\,nm$ are generated via type-II spontaneous parametric down conversion (SPDC) in a 2-mm BBO crystal, pumped by 392.5 nm pulses frequency-doubled by a 785-nm femtosecond laser. A 10-nm interference filter (IF) with the central wavelength of 785 nm is used. Orthogonally polarized photon pairs are separated by a polarizing beam-splitter (PBS) and coupled to single mode fibres (SMF). A delay line (DL) is inserted to control the temporal superposition of the photons. The polarization controller (PC1, quarter wave plate (QWP) + half wave plate (HWP) + QWP + Glan prism) is used to control the polarization of photons output from the fibres. A beam-splitter (BS) is employed for alignment of the two polarized beams. An input objective (× 50, NA = 0.4) and an oil objective (× 100, NA = 1.32) are used to excite the waveguide modes in the HW system and to collect the light from the sample, respectively. A triangular reflector is used to split the photons from the two outputs. After selected by polarization controller (PC2, HWP + Glan prism), the output photons are transmitted to the silicon avalanche photodiodes (APD) through multimode fibres (MMF) and coincidence measurements are made at the single photon counting modules (SPCM).

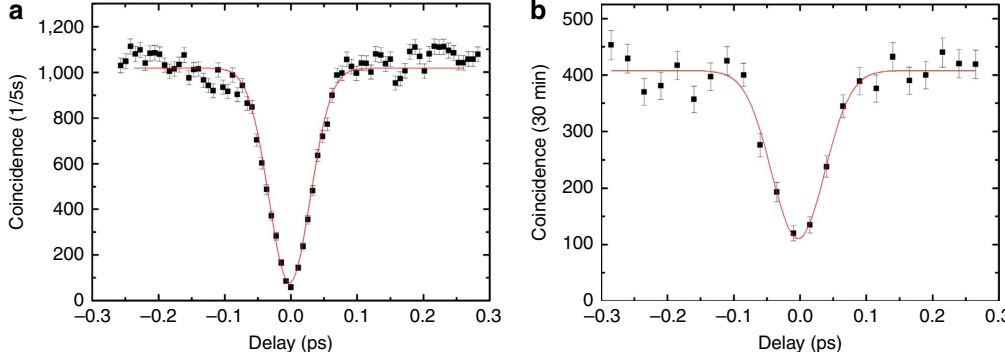

**Figure 3 | The HOM interference results.** (a) The HOM interference pattern of the photon pairs of the source. (b) The HOM interference pattern of the SPP at the HW-based PDBS. Black dots are data and the red lines correspond to fitting curves. Error bars are drawn to represent one standard deviation from the Poisson distribution.

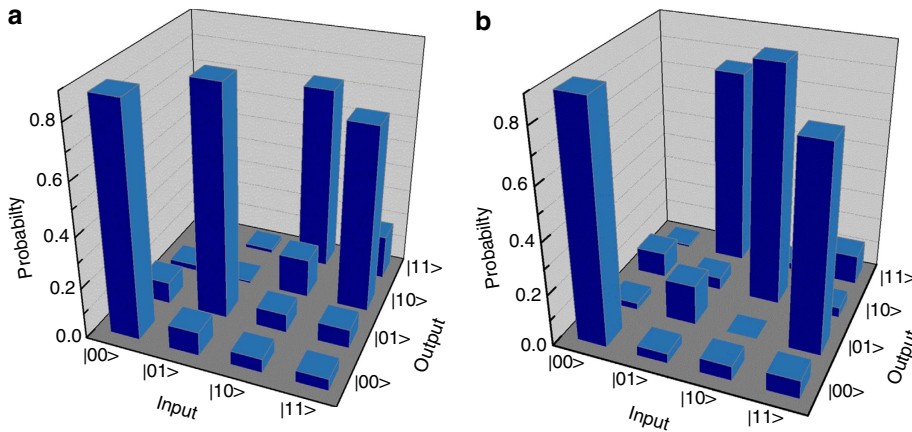

**Figure 4 | Characterization of the integrated gate.** Measured operation probabilities of the CNOT gate on the computational **ZZ** (**a**) and the **XX** (**b**) basis, respectively.

We then carry out the measurement of the operation of the CNOT gate. We first consider the **ZZ** basis defined as $|0_{zz}\rangle_c \equiv |s\rangle_c$, $|1_{zz}\rangle_c \equiv |p\rangle_c$ for the control qubit, and $|0_{zz}\rangle_t \equiv |D\rangle_t = (|s\rangle_t + |p\rangle_t)/\sqrt{2}$, $|1_{zz}\rangle_t \equiv |A\rangle_t = (|s\rangle_t - |p\rangle_t)/\sqrt{2}$ for the target qubit. Note that we define the control and target qubits in different basis as the gate constructed in this work is, at its core, a controlled-Z gate for $s$- and $p$-polarization, which requires two Hadamard operations on the input and output of the target qubit to become to a CNOT gate. Here, these Hadamard operations are achieved by a basis redefinition of the target qubit. Under post-selection, that is, a successful twofold coincidence measurement on the two output ports, the CNOT gate succeeds with a probability of 1/9 (refs 2,3). To characterize the operation of this gate, we measured the output of the gate for each of the four possible input states: $|00_{zz}\rangle_{ct}$, $|01_{zz}\rangle_{ct}$, $|10_{zz}\rangle_{ct}$ and $|11_{zz}\rangle_{ct}$, prepared by the polarization controller 1. The measured results for input–output operation probabilities, normalized by the sum of all coincidence counts obtained for each of the respective input states, is presented in Fig. 4a. The four correct output states corresponding to four input states are evident with high probabilities. Here, the probabilities of $88.3 \pm 8.3\%$ and $88.7 \pm 8.4\%$ for the input states $|00_{zz}\rangle_{ct}$ and $|01_{zz}\rangle_{ct}$, respectively, are higher than those of $72.1 \pm 7.5\%$ and $71.9 \pm 7.3\%$ for $|10_{zz}\rangle_{ct}$ and $|11_{zz}\rangle_{ct}$, respectively. The reason lies in that the non-classical interference relying on the overlap on the PDBS is required for the two latter input states, whereas it is not required for the two former input states. Some other non-zero probabilities can be attributable to the crosstalk between two polarizations from the scattering of the TM mode (SPP) into the TE mode in addition to the inaccuracy from the transmittance and reflectance of the PDBS. The average operation fidelity of the gate can then be obtained as $F_{zz} = 80.3 \pm 7.9\%$. These results show that this HW-based gate can well present the quantum CNOT function.

We then consider the complementary diagonal **XX** basis given by $|0_{xx}\rangle_c \equiv |D\rangle_c = (|s\rangle_c + |p\rangle_c)/\sqrt{2}$, $|1_{xx}\rangle_c \equiv |A\rangle_c = (|s\rangle_c - |p\rangle_c)/\sqrt{2}$ for the control qubit, and $|0_{xx}\rangle_t \equiv |s\rangle_t$, $|1_{xx}\rangle_t \equiv |p\rangle_t$ for the target qubit. As shown in Fig. 4b, the operation of the gate presents the correct output states $|00_{xx}\rangle_{ct}$, $|11_{xx}\rangle_{ct}$, $|10_{xx}\rangle_{ct}$, $|01_{xx}\rangle_{ct}$ corresponding to the input states $|00_{xx}\rangle_{ct}$, $|01_{xx}\rangle_{ct}$, $|10_{xx}\rangle_{ct}$, $|11_{xx}\rangle_{ct}$, respectively, with high probabilities given by $89.9 \pm 9.1\%$, $90.6 \pm 9.2\%$, $75.7 \pm 6.1\%$, $77.6 \pm 6.5\%$, respectively. The average operation fidelity can be computed as $F_{xx} = 83.5 \pm 7.7\%$. Then, the upper and lower bounds for the quantum process fidelity $F_{process}$ of the gate can be obtained via $F_{xx} + F_{zz} - 1 \leq F_{process} \leq \min[F_{xx}, F_{zz}]$[7]. Thus, the process fidelity of the HW-based quantum CNOT gate is $63.8 \pm 7.8\% \leq F_{process} \leq 80.3 \pm 7.9\%$.

The CNOT gate can be used as an entangling gate, which produces a two-qubit entangled output state from a separable input state. The lower bound of the process fidelity also defines a lower bound of the entanglement capability of the gate, as the fidelity of entanglement generation is at least equal to the process fidelity[7]. In terms of the concurrence $C$ that the gate can generate from the product state inputs, the minimal entanglement capability is given by $C \geq 2F_{process} - 1$ (ref. 7). As our experimental results show that the minimal process fidelity of the gate is 0.638, the lower bound of the entanglement capability can be $C \geq 0.276$ consequently. We also produced a post-selected polarization entanglement from a separable input state (see Supplementary Note 3 and Supplementary Fig. 3 for details). The high visibility also presents the good entangling function of the gate.

## Discussion

We finally discuss the integratabiltiy of this HW-based quantum logic gate. It seems that integratabtliy of our gate suffers from the low photon counting rates due to high losses of the plasmonic component. The high loss also reduces the signal/noise of the quantum measurement results, which will affect the performance of the HW-based quantum logic gate. However, in the matter of fact, the loss is mainly brought from the conversion processes between the spatial light and waveguide modes (SPP and TE mode) at the input and output couplers of the PDBS. Such coupling loss could be reduced by improving the measurement setup and coupling approach after this proof-of-principle work, such as using tapered nano-fibre coupling[20] or single-mode fibre coupling[21]. Furthermore, in the future integrated quantum processor containing on-chip sources and detectors, such conversion processing is not necessary and the losses of conversion processes can thus be further reduced. At this time, the attenuation function of the output couplers for TE mode can easily be fulfilled by some alternative small structures, such as gratings milled on the loaded dielectric layer. In addition, considering the wavelength of 785 nm in our work, the theoretically predicted propagating length in dielectric-loaded SPP waveguide is more than 100 μm. It is sufficient for accommodating multiple logical elements in creating sophisticated quantum information processing system. For the telecom wavelength (around 1,550 nm), the loss can be further reduced, leading to the millimetre propagation length and a larger scale chip[22]. Therefore, the signal/noise ratio of the quantum measurement results will also increase. Besides, the phase-damping

decoherence resulting from the index difference of the SPP and TE mode may also affect the integratability. As we discuss in the Supplementary Note 4 and Supplementary Fig. 4, the coherence length can be further increased using a narrow-band photon-pair source, which meets the scalability requirement of the quantum chip. Consequently, we believe that, with the advanced nano-fabrication techniques[23], the functional elements based on HW system can be conveniently integrated onto a chip to realize diverse logical functions or algorithms, all done within an ultra-compact footprint.

## Methods

**Polarization-encoded CNOT gate.** The CNOT gate based on polarization-encoding is schematically shown in Fig. 1a. The core part of the gate is a PDBS denoted as $PDBS_0$ that perfectly transmits the TE polarized light and allows for 1/3 (2/3) transmission (reflection) of the TM polarized light. In practice, when the PDBS is leaky, the CNOT gate still works if the transmittance ($T$) and reflectance ($R$) for the TE and TM polarized lights satisfy $T_{TE} = \xi$, $R_{TE} = 0$ and $T_{TM} = \xi/3$, $R_{TM} = 2\xi/3$, respectively. $\xi$ is related to the total loss including the conversion and propagation losses and so on. Auxiliary PDBSs (PDBSa), with $T_{TE}/T_{TM} = 1:2$, are employed to balance the contributions from the two polarizations. For the input and the extraction of on-chip signals, the construction of a single CNOT gate calls for the input and output couplers (Fig. 1a) as used in this work. Here, $\xi$ was estimated to be ∼1% from the experimental data in Fig. 1d–f.

**Experimental setup.** The experimental setup can be divided into three parts. The first part is the source: A 2.5 W pulsed 785 nm femtosecond laser from a Ti: Sapphire oscillator (Tsunami, Spectra-Physics Lasers) is frequency-doubled using a 2-mm-thick BBO crystal to obtain 392.5 nm pulses. The photon pairs with wavelength of 785 nm are then generated via spontaneous parametric down conversion in a 2-mm type-II BBO crystal pumped by these pulses. The photon pairs are filtered by a 10-nm interference filter with the central wavelength of 785 nm. The orthogonally polarized photon pairs are separated by a polarizing beam splitter and coupled to single-mode fibres. A delay line is inserted to control the temporal superposition of the photons. The polarization controller 1 (quarter wave plate + half wave plate + quarter wave plate + Glan prism) is used to compensate polarization transformations in the single-mode fibres and perform polarization unitary operations. The second part is the microscopy system, composed of the CNOT gate sample, an input objective ( × 50, numerical aperture (NA) = 0.4) used to excite different waveguide modes in the hybrid system, and an oil objective ( × 100, NA = 1.32) employed to collect the scattering light from the output slits of the gate. The final part comprises the collection and analysis apparatus, where photons scattering from the two outputs are separated by a triangular reflector placed at the image plane behind the oil objective. After selected by polarization controller 2 (half wave plate + Glan prism), the output photons are then delivered to the silicon avalanche photodiodes (Perkin Elmer) through multimode fibres and coincidence measurements are made at the single photon counting modules (SPCM, Becker & Hickl DPC-230).

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

## Acknowledgements

This work was supported by the National Key Projects of Basic Researches in China (No. 2012CB921501, No. 2012CB921802 and No. 2012CB933501), the National Natural Science Foundation of China (Nos. 11322439, 91321312, 11321063, 11422438 and 11474050). T.L. acknowledges the support of the Dengfeng Project B of Nanjing University and the PAPD project of Jiangsu Higher Education Institutions, C.S. acknowledges the support of State Administration of Foreign Experts Affairs "High-end foreign experts project" (No. GDW20153200149). We acknowledge Professor Xiaosong Ma and Professor Edna Cheung for their helpful discussion.

## Author contributions

S.M.W. and Y.X.G. conceived the idea and designed the experiments. Q.Q.C. and L.L. carried out the device fabrication. S.M.W. and Q.Q.C. did characterizations. S.M.W., Y.X.G., T.L. and C.S. analysed the results. S.M.W., T.L. and S.N.Z. supervised the work. All authors discussed the results and commented on the manuscript.
