## [Peer Review file · Nature Communications]

Transferred manuscripts:

Reviewers' Comments:

Reviewer #1 (Remarks to the Author)

The paper by Wang et al. submitted to Nature Communications is a revised version of the manuscript that I have refereed for another Nature journal. In my previous review I had already expressed my appreciation for this work, but raised several general and technical issues. I have carefully read the authors' replies to my previous comments and I think that they have suitably addressed most of the points. The only aspect still missing regards the device losses. I would recommend that the authors provide some numbers that could be a benchmark for further studies and a stimulus for improvement to the researchers in the field. Having considered the authors' responses and the different journal where the manuscript is now submitted, I would support publication on Nature Communications.

Reviewer #2 (Remarks to the Author)

The authors present fabrication and optical characterization of quantum CNOT gate structures. The complete CNOT operation of the gates has been realized on a small footprint of $14\ \mu\text{m} \times 14\ \mu\text{m}$. The input and output grating structures as well as the key element, the polarization dependent beam splitter (PDBS), have been fabricated by focused ion beam milling. The working principle of the PDBS relies on the different mode profiles of TE and TM modes in the waveguiding structure. All experiments have been performed with great care, and proper CNOT operation has been demonstrated. The results are novel and sound, the manuscript is well written, and references are adequately cited. The findings are of interest and importance for the photonics and quantum computing society. After the manuscript has been improved according to the reviewers' comments I recommend this contribution for publication.

To Reviewer #1:

The paper by Wang et al. submitted to Nature Communications is a revised version of the manuscript that I have refereed for another Nature journal. In my previous review I had already expressed my appreciation for this work, but raised several general and technical issues. I have carefully read the authors' replies to my previous comments and I think that they have suitably addressed most of the points. The only aspect still missing regards the device losses. I would recommend that the authors provide some numbers that could be a benchmark for further studies and a stimulus for improvement to the researchers in the field. Having considered the authors' responses and the different journal where the manuscript is now submitted, I would support publication on Nature Communications.

We thank the reviewer for the positive comment on our revised manuscript. As for the loss problem, we have added some discussion on it in the discussion part of the manuscript.

"The high loss also reduces the signal/noise of the quantum measurement results, which will affect the performance of the HW-based quantum logic gate."

To Reviewer #2:

The authors present fabrication and optical characterization of quantum CNOT gate structures. The complete CNOT operation of the gates has been realized on a small footprint of $14\ \mu\text{m} \times 14\ \mu\text{m}$. The input and output grating structures as well as the key element, the polarization dependent beam splitter (PDBS), have been fabricated by focused ion beam milling. The working principle of the PDBS relies on the different mode profiles of TE and TM modes in the waveguiding structure. All experiments have been performed with great care, and proper CNOT operation has been demonstrated. The results are novel and sound, the manuscript is well written, and references are adequately cited. The findings are of interest and importance for the photonics and quantum computing society. After the manuscript has been improved according to the reviewers' comments I recommend this contribution for publication.

We are very grateful to the reviewer's positive comment on our revised manuscript.